# Microbiota, Fiber, and NAFLD: Is There Any Connection?

**DOI:** 10.3390/nu12103100

**Published:** 2020-10-12

**Authors:** Alejandra Pérez-Montes de Oca, María Teresa Julián, Analía Ramos, Manel Puig-Domingo, Nuria Alonso

**Affiliations:** Endocrine and Nutrition Service, Germans Trias i Pujol Hospital and Research Institute, Universitat Autònoma de Barcelona, 08916 Badalona, Spain; alec148@gmail.com (A.P.-M.d.O.); mtjulian.germanstrias@gencat.cat (M.T.J.); analiaemilceramos@gmail.com (A.R.); mpuigd@igtp.cat (M.P.-D.)

**Keywords:** NAFLD, NASH, microbiota, fiber, prebiotics, diet

## Abstract

Gut microbiota can contribute to the development and progression of non-alcoholic fatty liver disease (NAFLD). In fact, some specific changes of gut microbiota are observed in patients in what is called dysbiota. There has been a lot of investigation by using a variety of interventions, including diet, showing the possibility to modify components of gastrointestinal dysbiota towards healthy and multivariate microbiota to restore physiologic status. One of the main focuses has been dietary fiber (DF), in which most of its variants are prebiotics. The highest effective treatment for NAFLD is, so far, weight loss achieved by caloric restriction. DF supplementation with oligofructose facilitates weight loss, enhances the production of beneficial metabolites, decreases some pathogenic bacteria population by increasing *Bifidobacteria*, and has effects on intestinal barrier permeability. DF use has been associated with improvement in diverse metabolic diseases, including NAFLD, by modifying gut microbiota. Additionally, it has been shown that a higher insoluble fiber consumption (≥7.5 g/day) revealed improvements in 3 different scores of liver fibrosis. Further research is needed, but given the evidence available, it is reasonable to prescribe its consumption in early stages of NAFLD in order to prevent disease progression.

## 1. Introduction

Non-alcoholic fatty liver disease (NAFLD) is rapidly becoming one of the most important causes of liver disease and its global prevalence is currently estimated at 24% [1]. The incidence of NAFLD has grown worldwide, in parallel to the obesity pandemic. NAFLD is strongly associated with obesity and other metabolic disorders such as type 2 diabetes (T2D) and dyslipidemia [2]. Along with these comorbidities, the most common cause of death for these patients is cardiovascular disease (CVD) [3]. They also have a high risk of liver-related morbidity and mortality [1]. Around 20–30% of patients with NAFLD will develop non-alcoholic steatohepatitis (NASH) that can lead to progressive liver damage, cirrhosis, and hepatocellular carcinoma [4]. The pathogenesis of NAFLD has not been fully elucidated. The most widely supported theory implicates insulin resistance as the key mechanism with an alteration in hepatic lipid homeostasis, mitochondrial dysfunction, and lipotoxicity [5]. It is well known that not only the liver plays a role, but also the pancreas, the stomach, the adipose tissue, the muscle, and more interesting the intestines and their microbiota [4].

NAFLD diagnosis requires demonstration of hepatic steatosis by imaging or biopsy, exclusion of significant alcohol consumption and other secondary causes, but liver biopsy remains as the gold standard for histological evaluation [6]. Currently, effective treatment for NASH is limited and drugs potentially useful are not yet completely established. Lifestyle interventions are recommended and mandatory by the American and European guidelines [3,6], while studies have shown that diet is one of the most important and promising treatments leading to the prevention and reversal of fibrosis, even in advanced stages [7,8]. What diet and whether the quantity is better than quality remains to be the focus of study. Evidence suggests that lowering caloric intake by at least 30% or by approximately 750–1000 Kcal/day results in an improvement in insulin resistance and hepatic steatosis [9]; and 7–10% of weight loss can lead to steatosis resolution and more importantly regression of fibrosis [6,7,8]. Even though studies mention that a hypocaloric diet is the most important component beyond quality or composition, a moderate restriction of carbohydrates and emphasis on high fiber and monounsaturated fatty acids seem to be a reasonable option [5]. Fiber consumption may play an important role in NAFLD, not only as a nutritional plan component that can help to lose weight but also as part of its pathogenesis involving intestine microbiota.

## 2. Intestinal Microbiota

The gastrointestinal tract (GIT) microbiota is one of the most densely populated microbial communities on earth [10,11]. It includes bacteria, fungi, viruses, and archaea, though the first ones are the overwhelming majority [12]. Its distribution is heterogeneous, being the large intestine the predominant location and the main site where the fermentation process occurs. The intestinal microbiota has been described as a “virtual organ” due to the different functions it performs. These include pathogenic defense, energy homeostasis, immune development, and an essential role in physiologic digestive function [12,13].

The gut microbiota that deviates from the ‘healthy’ status in terms of diversity and functionality is called dysbiotic. It has been associated with a lot of pathologies including inflammatory bowel diseases, obesity and T2D [14,15]. Alterations in gastrointestinal microbiota lead to intestinal barrier dysfunction through several mechanisms including a “leaky gut”. This happens when proteins that are responsible for tight junctions such as claudins, zonulin, and occludin, are compromised by changing its distribution and allowing a higher intestinal permeability to bacterial components [16,17,18]. These phenomena promote metabolic endotoxemia and contribute to the development of a chronic low-grade inflammatory state in both the adipose tissue and the liver [19,20]. Moreover, the integrity of the intestinal barrier could be disrupted by microbial metabolites such as ethanol, that has an increased production in a dysbiotic gut ecosystem, and other volatile organic compounds, leading to further greater liver injury in the setting of amplified hepatic fat accumulation [16]. Therefore, gut–liver axis derangement such as gastrointestinal dysbiosis and production of inflammatory molecules, among others lipopolysaccharide, might modulate the progression of NAFLD by promoting bacteria/bacterial product translocation into portal circulation and activation of inflammation via toll-like receptors signaling in hepatocytes [4,21].

The intestinal microbiota is different from host to host. Recent studies have identified gut microbes associated with potential beneficial outcomes (e.g., *Bifidobacterium, Lactobacillus, Faecalibacterium, Roseburia, Ruminococcus, Bacteroides* sp.) and potential harmful outcomes (e.g., *Clostridium, Enterobacter, Enterococcus* sp.). The beneficial outcomes include anti-inflammatory effects in the gut and favorable action in metabolic parameters [13]. Two studies [22,23] comparing NAFLD patients with healthy controls found an increased abundance of the *Lactobacillus* genus, and a decrease in the family *Ruminococcaceae* in NAFLD patients. Another study comparing NAFLD, NASH and healthy subjects described a decreased percentage of Bacteroides in NASH patients [24]. Moreover, a study comparing the microbiota composition in children with NASH vs. healthy and obese children found a gradual rise in *Proteobacteria* [25]. Furthermore, a cross-sectional study comparing the gut microbiome of obese and lean patients with or without NASH found that, in comparison with healthy subjects, lean NASH patients showed abundance of *Faecalibacterium*, and *Ruminococcus* sp., while obese NASH patients were enriched with Lactobacillus sp. More important, liver fibrosis ≥ F2 was associated with an increase of *Lactobacillus* sp. [26]. Interestingly, a study by Zhu based on the characterization of gut microbiomes in NASH patients found a higher abundance of alcohol-producing bacteria when compared to obese or healthy individuals with normal liver function. Specifically, *Escherichia* from the *Proteobacteria* phylum was significantly elevated in NASH patients. This study also described an increased blood alcohol concentration with the same pattern. This result suggests that microbes rich in ethanol-producing (*Bacteroides, Bifidobacterium,* and *Clostridium* sp.) may be a risk factor in driving the disease progression from obesity/NAFLD to NASH [27]. This theory is supported by other experimental and human studies where bacteria such as *Klebsiella pneumoniae* and bacteria from the *Lactobacillus* genus, both ethanol-producing bacteria, were involved in the pathogenesis of NAFLD [28,29].

To sum up, some diseases can modify gut microbiota, but its shaping is also influenced by many other factors and characteristics such as age, genetics, medications and, more relevant, diet. There has been a lot of investigation about how a variety of interventions, including nutritional plan, can affect and modify components of gut microbiota and have subsequent consequences for health status. One of the main focuses has been dietary fiber (DF) [10,11,13].

## 3. Dietary Fiber and Prebiotics

A variety of definitions of DF have been promulgated by scientific and regulatory agencies worldwide. The EU regulation on the provision of food information to consumers [30] defines fiber as “carbohydrate polymers with three or more monomeric units, which are neither digested nor absorbed in the human small intestine. DF has been categorized as follow: (i) edible carbohydrate polymers naturally occurring in the food as consumed, (ii) edible carbohydrate polymers obtained from food raw material by physical, enzymatic or chemical means and which have a beneficial physiological effect demonstrated by generally accepted scientific evidence, and (iii) edible synthetic carbohydrate polymers which have a beneficial physiological effect demonstrated by generally accepted scientific evidence”. DF represents the major non-digestible component in most diets and it exerts a physiological influence throughout the digestive tract from the modulation of digestion processes to acting as a prime substrate for microbial fermentation [12]. Particle size and shape, viscosity, as well as extent/rate of fermentation can significantly affect these different functions in the GIT [11].

There are a lot of classifications of DF. The most common one for human nutrition divides it into two subgroups based on its solubility in water: soluble vs. insoluble [12]. Although the World Health Organization (WHO) in 1998 proposed to no longer use this classification [30], it is useful as a predictor of its water-holding capacity, viscosity and degree of fermentation by GIT bacteria from a physicochemical point of view. Insoluble fibers such as cellulose usually found in bran, legumes, and nuts are generally poorly fermented by intestinal microbiota, but they boost gut transit rate and reduce the amount of time available for colonic bacterial fermentation of undigested foodstuff. On the other side, soluble fibers such as pectin and xyloglucans are highly fermentable and can be located in whole grains (e.g., oats and barley [β-glucan]) and fruits (e.g., apples [pectin]). Inulin, resistant maltodextrins, resistant starch, polydextrose and soluble corn fiber are others soluble DF that are readily fermented by gastrointestinal microbiota [11,12,16]. This fermentation process is one of the main benefits of DF, modulating intestinal microbiota, affecting its diversity and function and the by-products of the process itself [12]. In addition, the degradation of prebiotics provides short-chain fatty acids that fulfill a protective role for colonocytes helping to maintain the proper structure and function of the intestinal barrier [16].

Interestingly, some DFs are prebiotics. Prebiotics are a group of nutrients that are degraded by gut microbiota. The exact definition has changed over the time and in 2010 it included a focus on its functionality: “a selectively fermented ingredient that results in specific changes in the composition and/or activity of the gastrointestinal microbiota, thus conferring benefits upon host health” [31]. There are many types of prebiotics, the most important ones are fructooligosaccharides (FOS) and galactooligosaccharides (GOS). The majority of them can be classified as DF but not all fibers can be listed as prebiotic. They naturally exist in different dietary food products, including asparagus, garlic, onion, Jerusalem artichoke, wheat, honey, banana, tomato, soybean, human and cow’s milk, peas, beans, among others [11,17,31]. Moreover, nowadays they have been added as ingredients to many common food products such as bread and breakfast cereal [32].

Prebiotics have different well-known favorable outcomes. Production of beneficial metabolites (inulin, oligofructose), augmentation of calcium absorption (inulin, oligofructose, GOS), improvement in allergy risk (FOS/GOS), enhancement of the immune system (oligofructose), effects on gut barrier permeability (oligofructose), and decline in pathogenic bacteria population by expanding *Bifidobacteria* (inulin, oligofructose, GOS) are some of the most known benefits [17].

Several studies have found a relation between DF and prebiotic intake and a growing variety of health-beneficial bacteria. A high-fiber diet enlarges the abundance of *Bifidobacterium* and reduces the ratio of Firmicutes/Bacteroidetes in humans and experimental animals [13]. A randomized control trial (RCT) comparing the effects of a high fiber diet vs. calorie restricted diet in the modulation of gut microbiome dysbioses in T2D patients found a higher fecal abundance of several beneficial microbiota such as *Roseburia* sp., *Fecalibacterium* sp., and *Bacteroides* sp. in both groups. On the other hand, the fiber-rich diet group had a decreased of pro-inflammatory bacteria, specifically of *Collinsella* and *Streptococcus* sp. [33]. Furthermore, a dietary intervention comparing the effects of a diet enriched with two different types of fiber-arabinoxylan and resistant starch type 2- and a low fiber diet found a greater abundance of *Bifidobacterium* sp. and lowered microbial diversity in the fiber supplemented group [34]. Moreover, RCTs have proved that in healthy adults the consumption of GOS—1.5 to 10 g/day- for up to 12 weeks rose the fecal level of *Bifidobacterium* [11,13]. Other studies have shown that supplementation with GOS, inulin, and oligofructose has resulted in an abundance of *Bifidobacterium, Lactobacillus,* and *Faecalibacterium* sp. [13] (Table 1).

## 4. Dietary Fiber and NAFLD

The health impact of DF, and more recently prebiotics, has been extensively reviewed and accepted worldwide. As mentioned before, they have been linked to different beneficial outcomes [17]. More recently, their use has been associated with the enhancement in diverse metabolic diseases, including NAFLD [35].

As previously described, the most effective treatment for NAFLD is weight loss achieved by caloric restriction [8]. It is known that fiber supplementation reduces the frequency of eating by intensifying satiety through the stimulation of the anorexigenic hormones and suppression of the orexigenic hormone ghrelin. This beneficial effect, in addition to its low energy density, has linked them to weight reduction [36,37]. A randomized control trial (RCT) showed that, independent of other lifestyle changes, fiber supplementation with oligofructose has the potential to promote weight loss and improve glucose regulation in overweight adults compared to placebo [37].

In addition, some prebiotics such as inulin have been associated with reduced body weight or attenuated weight gain [36,38]. Beyond weight loss, animal studies suggest that dietary supplementation with prebiotic can have a positive effect on NAFLD by modifying gut microbiota, reducing body fat, and bettering glucose regulation [35]. In humans, an RCT found that increased fiber intake (soluble and insoluble), from 19 g/day to the 29 g/day, reduced serum zonulin concentration, decreased liver enzymatic activity, and enhanced hepatic steatosis in patients with NAFLD, possibly by modifying intestinal permeability [16]. In addition, a recent study established a relationship between the degree of liver fibrosis measured by non-invasive assessments and fiber intake. This study found that a higher insoluble fiber consumption (≥7.5 g/day) showed improvements in three different scores of liver fibrosis (fatty liver index, hepatic steatosis index, and NAFLD liver fat score), while significant amelioration in hepatic enzymes were observed as a result of fruit fiber consumption (≥8.8 g/day) [39]. Furthermore, a 12-week comparison between a commercially available formula diet supplemented with oats fibers versus a comparably restricted nutritional program found that both dietetic interventions were similarly effective regarding weight loss, but the diet supplemented with oats fibers was more efficient regarding the reduction of intrahepatic lipid content detected by hepatorenal index [40]. Other smaller studies and reviews, both in animals and humans, have found a positive association between a specific DF and NAFLD [41,42,43] (Table 2). High food fibers that are recommended and discouraged in NAFLD are shown in Table 3.

A “high-quality healthy diet” has been proposed to improve hepatic steatosis and metabolic dysfunction in patients with NAFLD, independent of caloric restriction and weight loss [44]. This nutritional program is supported by the idea that lower fiber intake is common in NAFLD [44,45] and is based on moderate to high carbohydrates intake (45–65% of total daily calories), low to moderate fat intake (below 30–35% of total calories with a high preference for healthy fat intake—monounsaturated fatty acids and omega-3 polyunsaturated fatty acids), protein intake (15–20% of total daily calories) and fiber intake increasing the consumption of fruits and vegetables, with more focus on prebiotic fiber. However, a high, fructose-rich diet in the form of added sugar is associated with more intestinal permeability, endotoxemia, higher hepatic TNF-α production, and lipid peroxidation, promoting hepatic steatosis and NAFLD [46].

This diet overlaps with the Mediterranean Diet (MD), which is the most recommended dietary pattern in NAFLD. Currently, clinical guidelines also recommend MD as the nutritional program of choice for NAFLD treatment [6]. In crossover comparisons between MD and a low-fat, high-carbohydrate diet, even without weight loss, MD reduces liver steatosis (assessed by magnetic resonance imaging) and improves insulin sensitivity in an insulin-resistant population with biopsy-proven NAFLD [47]. In different subgroups of the PREDIMED trial [48], an RCT aimed at evaluating the effect of MD on the primary prevention of CVD, supplemented with extra-virgin olive oils or nuts, it was found a reduced prevalence of hepatic steatosis and a delay in the progression of NAFLD [49].

## 5. Conclusions

Fiber intake positively influences NAFLD not only by promoting reduced calorie intake, but also by stimulating a healthy gut microbiota, therefore reducing the development of inflammation and liver injury. However, to date, no study has found regression of a more advanced stage of NAFLD, such as fibrosis, in patients with high intake of fiber or prebiotic supplementation. Further research is needed, but given the evidence it is reasonable to indicate its consumption in early stages of NAFLD in order to prevent disease progression.

## Figures and Tables

**Table 1 nutrients-12-03100-t001:** Potential beneficial gut microbiota changes according to modulations of dietary fiber and/or prebiotics [11,13].

Type of Diet, Fiber, and/or Prebiotic	Potential Beneficial Microbiota Changes	Type of Studies
High fiber diet	Increase in *Bifidobacterium, Prevotella, Lactobacillus, Faecalibacterium, Roseburia, Bacteroides* sp. Decrease in *Collinsella* and *Streptococcus* sp.	Cross-sectional, RCT
Resistant starch (type 4)	Increase in *Bifidobacterium* sp. Decrease in Firmicutes phylum	Cross-sectional
Resistant starch (type 3)	Increase in *Bifidobacterium* sp., *Ruminococcus (R. bromii)* and *Eubacterium (E. rectal)*	
Resistant starch (type 2)	Increase in *Bifidobacterium, Ruminococcus (R. bromii)* and *Eubacterium (E. rectal)*	
Arabinogalactan and Arabinoxylan	Increase in *Bifidobacterium, Bacteroides (B. ovatus)*, *Lactobacillus, Coprococcus* and *Lachnoclostridium* sp.	RCT, cross-sectional and in vitro
Galactooligosaccharides	Increase in *Bifidobacterium*, *Bacteroides* and *Lactobacillus* sp. Decrease in *Clostridium* sp.	RCTs and in vitro
Inulin	Increase in *Bifidobacterium* and *Faecalibacterium* sp. Decrease in *Enterococcus* sp.	RCTs and in vitro
Oligofructose	Increase in *Bifidobacterium* sp.	RCT

RCT: randomized control trial.

**Table 2 nutrients-12-03100-t002:** Published studies evaluating fiber consumption and NAFLD.

Author/Year	Type of Study	Dose, Treatment, and Follow Up.	Results
Daubioul et al., 2005 [35]	Randomized cross-sectional study	Daily ingestion of 16 g of oligofructose or maltodextrin (placebo) in biopsy-proven NASH patients for 8 weeks	Improvement in hepatic enzymes and insulin levels in NASH patients receiving a dietary supplementation with dietary fructans
Rocha et al., 2007 [50]	Cross-sectional study	Daily ingestion of 10 g of soluble fibers in patients with NAFLD during 3 months	After fiber supplementation, 75% of the patients presented normal liver enzymes
Bozzetto et al., 2012 [51]	RCT	Effects of qualitative dietary changes and exercise (CHO/fiber vs. MUFA diet) in obese/overweight patients with T2D during 8 weeks	Liver fat content decreased more in MUFA diets groups. High-fiber, low-glycemic index diet did not influence liver fat content
Cantero et al., 2017 [39]	RCT	Influence of two energy restricted diets (AHA diet vs. RESMENA diet) on non-invasive markers and scores of liver damage in obese patients for 6 months	In both dietary strategies, increased insoluble fiber consumption (≥7.5 g/day) showed improvements in 3 different scores of liver fibrosis (fatty liver index, hepatic steatosis index, and NAFLD liver fat score)
Krawczyk et al., 2018 [16]	RCT	Increased fiber intake from 19 g/day to the 29 g/day (soluble and insoluble) in patients with NAFLD for 6 months	Significant improvements in hepatic enzymes and of fatty liver status according to the Hamaguchi score. Decreased Zonulin concentration by nearly 90% and correlated with the amount of dietary fiber intake as well as the degree of fatty liver
Schweinlin et al., 2018 [40]	RCT	Comparison of a formula-based nutritional therapy enriched with oats fiber with a non-formula isocaloric therapy in obese patients for 12 weeks	Diet supplemented with oats fibers was more effective regarding the reduction of intrahepatic lipid content detected by hepatorenal index (1.1 ± 0.2 vs. 1.9 ± 0.3, *p* < 0.05)

RCT: randomized control trial, NASH: non-alcoholic steatohepatitis, NAFLD: non-alcoholic fatty liver disease, AHA: American Heart Association, RESMENA: Reduction of Metabolic Syndrome in Navarra, T2D: type 2 diabetes, CHO: carbohydrates, MUFA: monounsaturated fatty acids.

**Table 3 nutrients-12-03100-t003:** High food fibers that are recommended and discouraged in NAFLD.

Less Recommended	Most Recommended	
Corn	Onion	Tomato
Rice	Cereals	Soybean
Soft drinks	Garlic	Oat and barley
Fruit juices	Leeks	Seed plants
Honey	Asparagus	Wheat
Syrup	Mushrooms	Jerusalem artichoke

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
