# Peer review of "Microbiota, Fiber, and NAFLD: Is There Any Connection?"

_nutrients, 2020, doi:10.3390/nu12103100_

Round 1

Reviewer 1 Report

General comments

The manuscript by Perez-Montes de Oca and colleagues aimed to summarize the current evidence on the relationship between gut microbiota, fiber intake and NAFLD.

The topic is relevant and fit with the aims of the Journal.

Specific comments

Table 2. The authors missed some important RCTs. Please refer to Bozzetto et al. Nutrients, 2018 (doi: 10.3390/nu10070943) to find missing studies.

Figure 1. The figure is totally misleading and I did not get the meaning of the duplicate. I suggest to use boxes.

In addition, I totally disagree with the presence of wheat as less recommended item. Many studies have demonstrated that wheat fiber can be fermented with the consequent production of SCFAs. As an example,  Vetrani et al demonstrated that a whole wheat-based diet increased  propionate concentration and this increase associated with the improvement of insulin concentrations (Nutrition,  2016; doi: 10.1016/j.nut.2015.08.006.) Hyperinsulinemia is a risk factor for NAFLD. In addition, propionate  that has been associated with reduced liver steatosis, inflammation and insulin resistance  (Tilgh et al. Gut, 2016; doi: 10.1136/gutjnl-2016-312729).

Author Response

Comment 1. Table 2. The authors missed some important RCTs. Please refer to Bozzetto et al. Nutrients, 2018 (doi: 10.3390/nu10070943) to find missing studies.

  • Reviewer is right. According to the sugestion, we have revised Bozzetto manuscript. In the new version we have introduced Bozzetto RCT in the table 2 as recommended. The other important RCT mentioned in the article (reganding fiber consumption and NAFLD) was already cited (Daubioul et al.).

Comment 2 and 3. Figure 1. The figure is totally misleading and I did not get the meaning of the duplicate. I suggest to use boxes. In addition, I totally disagree with the presence of wheat as less recommended item. Many studies have demonstrated that wheat fiber can be fermented with the consequent production of SCFAs. As an example,  Vetrani et al demonstrated that a whole wheat-based diet increased  propionate concentration and this increase associated with the improvement of insulin concentrations (Nutrition,  2016; doi: 10.1016/j.nut.2015.08.006.) Hyperinsulinemia is a risk factor for NAFLD. In addition, propionate  that has been associated with reduced liver steatosis, inflammation and insulin resistance  (Tilgh et al. Gut, 2016; doi: 10.1136/gutjnl-2016-312729).

  • We thank reviewer for this comment. Following suggestions, we have now changed the figure and transformed it into a table (table 3). After reviewing the citations mention aboved, wheat has been moved from less recommended to most recommended high fiber food.

Reviewer 2 Report

In this review Alejandra Pérez-Montes de Oca et al., evaluated that fiber intake positively affects NAFLD promoting a healthy gut microbiota. The topic is very interesting, but there are a few points that should be further  clarified.:

  • The manuscript is not in the final version, please remove the corrections in the text.
  • In the “intestinal microbiota section” please show more clearly the specific gut bacteria mainly associated with NAFLD progression .
  • In the "Dietary fiber section" describe the main benefits of prebiotics, for example fructooligosaccharides and galactooligosaccharides. Furthermore, I believe it is necessary to briefly describe the main studies that have found a relationship between Df and the intake of prebiotics with the growth of beneficial bacteria.
  • The description in the text of figure 1 is missing.
  • Please correct figure 1

Author Response

Comment 1. The manuscript is not in the final version, please remove the corrections in the text.

  • We apologize for this error. Corrections have been removed and the manuscript is now in the final version.

Comment 2. In the “intestinal microbiota section” please show more clearly the specific gut bacteria mainly associated with NAFLD progression .

  • After rewieng the literature of NASH and gut microbiome, new information have been introduced in the text (intestinal microbiota section). It refeers mainly to the differences in gut microbiota between NAFLD/NASH patients vs healthy subjects (line 86-104)

Comment 3. In the "Dietary fiber section" describe the main benefits of prebiotics, for example fructooligosaccharides and galactooligosaccharides.

  • Following reviewer recomendation, we added a paragraf with health benefits and the specifcs prebiotics linked to them. (line 143-153)

Comment 4. Furthermore, I believe it is necessary to briefly describe the main studies that have found a relationship between Df and the intake of prebiotics with the growth of beneficial bacteria.

  • Following the rewier suggestion, more detail information about some of the studies described in table 1 (Potential beneficial gut microbiota changes according to modulations of dietary fiber and/or prebiotics) was included in the text. (line 154-167)

Comment 5. The description in the text of figure 1 is missing.

  • Reviewer is right. Information regarding the figure (now Table 3) was introduced in the main text.

Comment 6. Please correct figure 1

  • We changed the figure and transformed it into a table in this new version follong reviewer 1 suggestion.

Round 2

Reviewer 1 Report

The manuscript has improved after the revision. 

Reviewer 2 Report

Accept in present form

This manuscript is a resubmission of an earlier submission. The following is a list of the peer review reports and author responses from that submission.